# Socioeconomic inequalities in abdominal obesity among Peruvian adults

**Marioli Y. Farro-Maldonado**[1], **Glenda Gutiérrez-Pérez**[1], **Akram Hernández-Vásquez**[2], **Antonio Barrenechea-Pulache**[1]*, **Marilina Santero**[3], **Carlos Rojas-Roque**[4], **Diego Azañedo**[5]

**1** Universidad Científica del Sur, Lima, Peru, **2** Vicerrectorado de Investigación, Centro de Excelencia en Investigaciones Económicas y Sociales en Salud, Universidad San Ignacio de Loyola, Lima, Peru, **3** Universitat Autònoma de Barcelona, Barcelona, Spain, **4** Universidad de Buenos Aires, Buenos Aires, Argentina, **5** Universidad Peruana Cayetano Heredia, Facultad de Ciencias y Filosofía, Lima, Peru

* abarrenechea@cientifica.edu.pe

**Data Availability Statement:** The database is freely accessible from the National Institute of Statistics and Informatics (INEI) website (http://iinei.inei.gob.

## Abstract

### Objectives

Abdominal obesity (AO) has become a public health issue due to its impact on health, society and the economy. The relationship between socioeconomic disparities and the prevalence of AO has yet to be studied in Peru. Thus, our aim was to analyze the socioeconomic inequalities in AO distribution defined using the International Diabetes Federation (IDF) cutoff points in Peruvian adults in 2018–2019.

### Methods

This was a cross-sectional study using data from the 2018–2019 Demographic and Family Health Survey (ENDES) of Peru. We analyzed a representative sample of 62,138 adults over 18 years of age of both sexes from urban and rural areas. Subjects were grouped into quintiles of the wealth to calculate a concentration curve and the Erreygers Concentration Index (ECI) in order to measure the inequality of AO distribution. Finally, we performed a decomposition analysis to evaluate the major determinants of inequalities.

### Results

The prevalence of AO among Peruvian adults was 73.8%, being higher among women than men (85.1% and 61.1% respectively, p < 0.001). Socioeconomic inequality in AO was more prominent among men (ECI = 0.342, standard error (SE) = 0.0065 vs. ECI = 0.082, SE = 0.0043). The factors that contributed most to inequality in the prevalence of AO for both sexes were having the highest wealth index (men 37.2%, women 45.6%, p < 0.001), a higher education (men 34.4%, women 41.4%, p < 0.001) and living in an urban setting (men 22.0%, women 57.5%, p < 0.001).

### Conclusions

In Peru the wealthy concentrate a greater percentage of AO. The inequality gap is greater among men, although AO is more prevalent among women. The variables that most

pe/microdatos/). The information can be obtained by entering the survey query tab and selecting the ENDES 2018-2019; data is obtained from modules #64, #65 and #414.

**Funding:** The authors received no specific funding for this work.

**Competing interests:** The authors have declared that no competing interests exist.

contributed to inequality were the wealth index, educational level and area of residence. There is a need for effective individual and community interventions to reduce these inequalities.

## Introduction

Over the last decades, the high prevalence of abdominal obesity (AO) has become a global public health issue with great social and economic impact. AO is a risk factor for multiple non-communicable diseases such as diabetes, cardiovascular disease, cancer [1] and has recently been identified as a predisposing factor for severe forms of COVID-19 [2]. This represents a high cost of treatment and its complications will exceed 5% of the annual health budget for the next 30 years [3]. In high-income countries such as the United States and Portugal, the prevalence of AO reaches 57.2% [4] and 50.5% [5] respectively, being even higher in upper-middle-income countries such as Mexico where it reaches 74.0% [6]. Using the International Diabetes Federation (IDF) cut-off points, the prevalence of AO in Peru in 2013 was 64.1% [7]. This disease threatens to overload the economic and resolutive capacity of health systems, particularly in Latin American countries in which budgets assigned to health are very limited, being of around 7.9% of the Gross Domestic Product (GDP) in 2018 compared to 16.9% in the United States [8].

The World Health Organization (WHO) recommends using the body mass index (BMI) to define obesity [9]. However, this parameter does not discriminate between muscle mass and lean mass [10]. The recommended BMI values were obtained in a Caucasian population in which the average height and body fat distribution is different from that found in other countries and may therefore underestimate the prevalence of overweight and obesity [11]. One study reported that 1 out of every 3 individuals with "normal weight" determined with the BMI had AO [12].Taking these results into account, waist circumference and the concept of AO is now being adopted in various studies [13, 14]. Waist circumference is easily applicable and correlated with the presence of visceral fat measured in tomographic studies [15]; it is recognized as a more accurate indicator of cardiovascular disease than the BMI [13]. Although in Peru, the Demographic and Family Health Survey (ENDES, acronym in Spanish) measures waist circumference, which is recognized by the Ministry of Health as part of the nutritional anthropometric assessment of adults, national reports and most local studies still consider the BMI as the main measure of obesity [16]. Nonetheless, this is controversial due to the low average height of the Peruvian population, being 165 cm for men and 153 cm for women [17].

Many researchers have reported socioeconomic variables influencing inequalities in the distribution of AO. In Indonesia, for example, the main determinants associated with inequalities were wealth status, occupational class, and educational level [18]. Currently, even with the information available, the influence of socioeconomic factors on the prevalence of AO in Peru has not been explored.

Therefore, this article aimed to analyze the socioeconomic inequalities in AO distribution, using the International Diabetes Federation (IDF) cut-off points for South and Central America in Peruvian adults using information from the 2018–2019 ENDES. The results of this study will help to better identify populations at risk of developing complications associated with AO and may be useful as a baseline for the elaboration of health policies aimed at the prevention of AO.

## Materials and methods

### Study population and design

Peru is a country divided into 24 sub-national administrative units, known as "administrative regions" and 1 constitutional province. The territorial area is 1,285,215.60 km2 and borders Ecuador, Colombia, Brazil, Bolivia and Chile. The total population in 2019 was 32,131,400 million people, being the 7th most populated country in America [19]. Peru can be divided into three natural regions: the coast, which concentrates 58% of the national population and many of the most developed cities including Lima, the capital [20]; the jungle, which is difficult to access due to the rugged terrain of the Amazon and whose population has insufficient access to basic services; and the highlands, the Andean area which presents the highest level of monetary poverty in the country. According to The World Bank the economy of Perú belongs to the upper middle income (gross national income per capita between $4,046 and $12,535) [21]. In 2018, 5.2% of the GDP was invested in health, being one of the lowest compared to other South American countries such as Colombia, Chile and Brazil [8].

This was cross-sectional study that used data from the 2018–2019 ENDES carried out by the National Institute of Statistics and Informatics (INEI, acronym in Spanish) of Peru. The ENDES is an annual survey, the objective of which is to obtain up-to-date information about the demographic dynamics and health condition of mothers, children younger than 5, and people older than 15 years of age who reside in Peru. It uses a two-stage, balanced, stratified and probabilistic sample, which is representative at national, administrative region and natural region levels. Each year studied had a sample size of 36,760 households, of which one individual 15 years of age or older was included in the survey. We used information compiled from both the household and the health questionnaires to carry out a secondary analysis to determine the prevalence and inequalities in the distribution of AO in adults. All measurements made during the survey were carried out by trained staff. Details about the procedures and measurement of waist circumference have been published by the INEI and can be found in the ENDES datasheet [22]. The sampling techniques and estimation of weighting factors can be consulted in the technical report [23].

In our study the units of analysis were individuals $\geq$ 18 years old residing in the selected sample of urban or rural households. We included 31,553 and 30,585 individuals from the 2018 and 2019 datasets, respectively. Both datasets were pooled to increase the power (n = 62,138).

### Variables

In our study, AO defined as a waist circumference $\geq$ 90 cm for men and $\geq$ 80 cm for women was the dependent variable. These cut-off points were recommended by the IDF for use in South and Central America [24]. The wealth index was the independent variable. This is a measure of household wealth constructed using the principal component analysis method which considers the availability of goods, services and housing characteristics [25]. In addition, to characterize the population, we grouped the wealth index into 5 quintiles (the first quintile being that with the lowest level of well-being and quintile five indicating the highest). AO was also analyzed according to the following population characteristics: 1) age group: 18–29 years, 30–59 years, 60 years or more; 2) marital status: never married, married or cohabiting and separated/divorced or widowed; 3) educational level: no formal school, primary, secondary, higher; 4) chronic disease: yes, no [reporting of at least one chronic condition including: hypertension, diabetes mellitus or depression]; 5) smoker: yes, no [having smoked during the last 30 days]; 6) area of residence: urban or rural; 7) altitude above sea level of the housing

conglomerate: 0–499 (meters above sea level), 500–1499 m.a.s.l., 1500–2999 m.a.s.l., and 3000 or more; and 8) natural region: jungle, mountain range, rest of coast, and Metropolitan Lima.

## Statistical analysis

We used weighted frequencies and their 95% confidence intervals to describe the socioeconomic characteristics of the study population and the prevalence of AO. Waist circumference was described using means and standard error (SE). The prevalence of AO was standardized according to the ages of the reference population indicated by the WHO [26]. Age standardization produces an age-adjusted prevalence, which is a weighted average, for each of the populations to be compared. Thus, standardization better represents the relative age distribution of the population. The prevalence of AO and the mean waist circumference were described according to sex.

To measure the socioeconomic inequality in the distribution of AO across the population, subjects were grouped into quintiles of wealth to calculate the concentration curve and the concentration index (CI). This curve has been used in other health indicators to describe the gradient related to socioeconomic inequality [27, 28] On the X-axis we plotted the cumulative percentage of the sample, ranked by the wealth index, and on the Y-axis we plotted the cumulative percentage of AO according to sex. A curve above the line of equality indicates a greater concentration of AO among the poor and vice versa. CI is a common method to measure income related inequality in health [27]. CI shows the covariance of the AO and the fractional rank of income distribution as:

$$CI = \frac{2}{\mu} cov_w \left[ y_{it} R_i^t \right] \qquad [a]$$

where $i$ is an individual, $y_i$ is AO (yes/no), $\mu$ is the mean of AO and R is the fractional rank in the income distribution. CI represents the concentration curve as a single number by summarizing the inequality weights at different points in the income distribution. It ranges between −1 and +1: negative values indicate that AO is concentrated among the poorest individuals, the same is true for the opposite results. One shortfall of the CI is that in the scenario of data contamination, the index is sensitive to extreme values at one or both tails of the distribution [29]. However, its main advantages are that it reflects the socioeconomic dimension to inequalities in health and the experiences of the entire population. The CI is sensitive to changes in the distribution of the population across socioeconomic groups, and, therefore, has been widely used to measure inequality within health economics.

Since AO is a binary variable, CI has the limitation that when the mean increases, the range of the possible values of the CI shrinks, tending to zero as the mean tends to one [31]. To solve this disadvantage, Erreygers introduced the Erreygers concentration index (ECI), a concentration index more compatible with a binary dependent variable [30, 31]. Mathematically, ECI can be expressed as:

$$E[h] = \frac{4\mu}{[b_n - a_n]} C[h] \qquad [b]$$

where $C[h]$ represents the standard CI, $\mu$ is the mean of AO in population and $b_n$ and $a_n$ are the upper and lower limits of AO. Following the Van Doorslaer methodology [32], decomposition analysis was performed to assess how much the independent variables contribute separately to socioeconomic inequality in AO. The decomposition was performed based on generalized linear models (GLM). In comparison to other approaches such as probit estimations or the ordinary least squares, GLM has shown to be the best choice when decomposing

inequalities using a binary outcome [33]. This study decomposes the inequality of abdominal obesity using the following equation.

$$ECI = 4 * (\Sigma_k(\beta_k^m \underline{x} k)CI_k + GCI_\varepsilon \qquad [c]$$

Where *ECI* Is the Erreygers concentration index, $\underline{x}k$ is the mean of the explanatory variables included in the decomposition (the socioeconomic and demographic factors), $\beta_k^m$ is the partial effect (*dy/dk*) evaluated at the sample means, $CI_k$ is the mean of the concentration index, and $GCI_\varepsilon$ is the generalized concentration index of the stochastic term of error. Eq [c] reflects that an explanatory variable contributes to the inequality in AO when this variable is correlated with AO and is not equally distributed across the wealth index. The contribution of the explanatory variable to the inequality in AO depends on the absolute value of the partial effect and the unequal distribution of the explanatory variable with respect to household income per-capita. A positive sign of the partial effect means that the explanatory variable contributes to an increase in the inequality observed, and vice versa [27]. All analyses were performed using Stata version 14.2.

Additionally, a sub-analysis of the prevalence of OA according to the sociodemographic characteristics of the population and the administrative region of residence was performed using the cut-off points recommended by the guidelines of the Third Adult Treatment Panel (ATP III) [34] and the Latin American Consortium of Studies in Obesity (LASO) [35] in which AO was defined as >102 cm in men, > 88 cm in women; and $\geq$ 97 cm in men, $\geq$ 94 cm in women, respectively.

## Ethical considerations

The Institutional Research Ethics Committee of the *Universidad Científica del Sur* (registration code: 560-2020-PRE15) approved the execution of this study.

## Results

### Population characteristics

After applying inclusion (adults over 18 years with waist circumference measurement) and exclusion criteria (individuals <18 years, pregnant, incomplete data), a total of 31,553 and 30,585 individuals were included from the 2018 and 2019 datasets, respectively (Fig 1). We included a total of 26,789 men and 35,349 women in the analysis.

Table 1 describes the socioeconomic characteristics of the population. More than half of the men and women ranged between 30–59 years of age and most were married or cohabiting. About 37.0% of men and 33.0% of women had a higher education. Most participants lived in the urban area (80.6%), a similar percentage of men and women lived in this area (80.3% and 80.9%, respectively, *p* = 0.092). Chronic diseases were more prevalently among men than women (24.6% vs. 20.8%, *p* < 0.001). Likewise, smoking was more prevalent among men than women (18.9% vs. 4.5%, *p* < 0.001) (Table 1).

### Mean waist circumference and prevalence of AO

The mean waist circumference was higher among men than women (93.5 cm vs. 92.3 cm, respectively, *p* < 0.001). Among men the mean waist circumference increased linearly across the categories of variables so that those with the highest wealth index, aged 60 or more, and with a higher education had the largest waist circumference. Meanwhile, among women the mean waist circumference tended to be higher among the middle categories; for example, women with a middle wealth index, those aged 30–59, and women with primary education

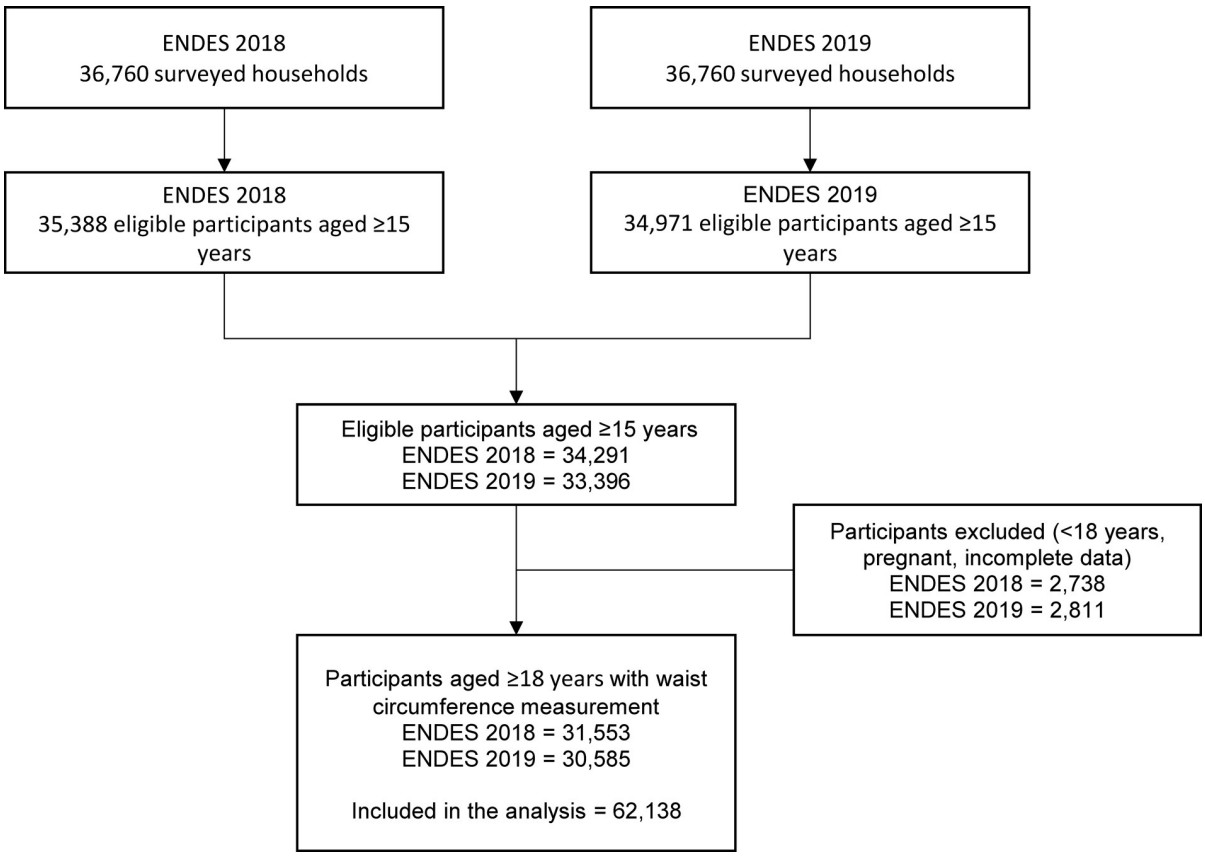

**Fig 1. Flowchart of the study population included from the 2018–2019 ENDES.**

had the highest waist circumference. In the case of chronic disease and altitude, in both sexes, individuals that reported having a chronic disease and who lived between 0–499 m.a.s.l. had the highest mean waist circumference (Table 2).

Regarding the prevalence of AO, women presented a higher prevalence than men both in general (85.1% vs. 61.1%, p < 0.001) and across all other independent variables. For women the highest prevalence of AO was concentrated among those from 30–59 years of age. Among individuals 18–29 years of age, we found a difference of 34.7 percentage points between the two sexes, favoring women. According to educational level, the prevalence of AO in men increased linearly with the level of education, while in women AO was concentrated among those with a primary level of education, maintaining a prevalence above 70% across the different educational levels. In both sexes the prevalence of AO was higher in metropolitan Lima and the rest of the coast; being 70.0% and 67.1% respectively in men and remaining above 80% across all the natural regions in women. We observed a lower prevalence of AO in men without chronic diseases compared to those who had comorbidities (55.8% vs. 78.9%, p < 0.001), while in women the rate of AO was high in both groups (83.5% vs. 93.1%, p < 0.001) (Table 2).

## Inequality in the distribution of abdominal obesity

The concentration curves for AO in men and women were both below the line of equity which shows that the cumulative percentage of AO was concentrated among the wealthiest individuals. The pro-rich orientation was more marked in men than in women. The ECI for AO in

**Table 1. Socioeconomic characteristics of a Peruvian adult population.**

| Characteristics | Overall n | Overall % (95% CI) | Men n | Men % (95% CI) | Women n | Women % (95% CI) | p-value[a] |
|---|---|---|---|---|---|---|---|
| Sample size | 62138 | 100 | 26789 | 100 | 35349 | 100 | |
| **Age groups, years** | | | | | | | |
| 18–29 | 17261 | 27.2 (26.7–27.8) | 6559 | 27.8 (27.0–28.7) | 10702 | 26.7 (26.0–27.4) | 0.038 |
| 30–59 | 35046 | 54.6 (54.0–55.2) | 15726 | 54.5 (53.6–55.5) | 19320 | 54.6 (53.8–55.4) | |
| 60 or more | 9831 | 18.2 (17.7–18.7) | 4504 | 17.6 (16.9–18.4) | 5327 | 18.7 (18.0–19.4) | |
| **Marital status** | | | | | | | |
| Never married | 8518 | 17.2 (16.7–17.7) | 4316 | 19.9 (19.1–20.7) | 4202 | 14.6 (14.0–15.3) | <0.001 |
| Married/Cohabiting | 43392 | 66.1 (65.4–66.7) | 19850 | 70.1 (69.2–71.0) | 23542 | 62.2 (61.3–63.0) | |
| Separated/Divorced/Widowed | 10228 | 16.8 (16.3–17.3) | 2623 | 10 (9.5–10.6) | 7605 | 23.2 (22.5–24.0) | |
| **Education level** | | | | | | | |
| No formal school | 3306 | 4 (3.8–4.3) | 537 | 1.5 (1.3–1.7) | 2769 | 6.5 (6.1–6.8) | <0.001 |
| Primary | 15698 | 20.6 (20.2–21.1) | 6358 | 18.1 (17.4–18.7) | 9340 | 23.1 (22.4–23.7) | |
| Secondary | 24696 | 40.1 (39.5–40.8) | 11636 | 43.4 (42.5–44.4) | 13060 | 36.9 (36.1–37.7) | |
| Higher | 18438 | 35.2 (34.6–35.9) | 8258 | 37 (36.0–38.0) | 10180 | 33.5 (32.7–34.4) | |
| **Wealth Index** | | | | | | | |
| Poorest | 19854 | 18.5 (18.1–18.9) | 8732 | 18.6 (18.0–19.2) | 11122 | 18.4 (17.8–18.9) | 0.064 |
| Poorer | 15551 | 20.8 (20.3–21.4) | 6726 | 21.4 (20.6–22.2) | 8825 | 20.3 (19.7–21.0) | |
| Middle | 11384 | 20.8 (20.2–21.3) | 4775 | 20.5 (19.8–21.3) | 6609 | 21 (20.3–21.7) | |
| Richer | 8738 | 20.1 (19.5–20.7) | 3749 | 20.3 (19.5–21.2) | 4989 | 19.9 (19.2–20.7) | |
| Richest | 6611 | 19.9 (19.2–20.5) | 2807 | 19.2 (18.3–20.2) | 3804 | 20.5 (19.7–21.3) | |
| **Natural regions** | | | | | | | |
| Jungle | 14302 | 12.1 (11.6–12.5) | 6318 | 12.5 (12.0–13.1) | 7984 | 11.6 (11.1–12.1) | 0.021 |
| Mountain Range | 22962 | 24.9 (24.2–25.6) | 9661 | 24.3 (23.5–25.2) | 13301 | 25.5 (24.7–26.3) | |
| Rest of Coast | 17705 | 25.7 (25.0–26.3) | 7610 | 25.5 (24.7–26.3) | 10095 | 25.8 (25.1–26.5) | |
| Metropolitan Lima | 7169 | 37.4 (36.7–38.1) | 3200 | 37.7 (36.6–38.8) | 3969 | 37.1 (36.2–38.0) | |
| **Area of residence** | | | | | | | |
| Rural | 21656 | 19.4 (19.0–19.8) | 9700 | 19.7 (19.2–20.3) | 11956 | 19.1 (18.6–19.5) | 0.092 |
| Urban | 40482 | 80.6 (80.2–81.0) | 17089 | 80.3 (79.7–80.8) | 23393 | 80.9 (80.5–81.4) | |
| **Altitude (meters above sea level)** | | | | | | | |
| 0–499 | 30259 | 65.7 (64.8–66.6) | 13182 | 66.2 (65.1–67.2) | 17077 | 65.3 (64.3–66.2) | 0.043 |
| 500–1499 | 7256 | 8 (7.3–8.8) | 3202 | 8.1 (7.3–8.9) | 4054 | 7.9 (7.1–8.7) | |
| 1500–2999 | 9663 | 11.7 (11.1–12.3) | 4145 | 11.6 (10.9–12.3) | 5518 | 11.7 (11.1–12.4) | |
| 3000 or more | 14960 | 14.6 (14.1–15.2) | 6260 | 14.1 (13.5–14.9) | 8700 | 15.1 (14.5–15.8) | |
| **Chronic disease** | | | | | | | |
| No | 50284 | 77.4 (76.8–77.9) | 20870 | 75.4 (74.5–76.2) | 29414 | 79.2 (78.5–79.9) | <0.001 |
| Yes | 11854 | 22.6 (22.1–23.2) | 5919 | 24.6 (23.8–25.5) | 5935 | 20.8 (20.1–21.5) | |
| **Smoker[b]** | | | | | | | |
| No | 55764 | 88.5 (88.0–88.9) | 21546 | 81.1 (80.3–81.8) | 34218 | 95.5 (95.1–95.9) | <0.001 |
| Yes | 6374 | 11.5 (11.1–12.0) | 5243 | 18.9 (18.2–19.7) | 1131 | 4.5 (4.1–4.9) | |

Weight specifications included the expansion factor and the ENDES sample specifications.

CI: confidence interval

[a]P-value for chi2 test of difference between men and women

[b]Smoked during the previous 30 days

**Table 2. Mean waist circumference and prevalence of abdominal obesity in men and women by socioeconomic characteristics.**

| Characteristics | Mean waist circumference (SE), cms | | | | Prevalence of abdominal obesity (%) | | | |
|---|---|---|---|---|---|---|---|---|
| | Men | P-value[a] | Women | P-value[a] | Men | P-value[b] | Women | P-value[b] |
| **Age-standardized sample[d]** | 93.5 (0.10) | | 92.3 (0.09) | <0.001[c] | 61.1 (60.3–61.9) | | 85.1 (84.6–85.7) | <0.001[c] |
| **Age groups, years** | | | | | | | | |
| 18–29 | 87.0 (0.21) | <0.001 | 86.6 (0.16) | <0.001 | 36.1 (34.4–37.9) | <0.001 | 70.8 (69.4–72.2) | <0.001 |
| 30–59 | 96.0 (0.14) | | 94.7 (0.13) | | 70.9 (69.9–71.9) | | 92.1 (91.5–92.7) | |
| 60 or more | 96.4 (0.24) | | 94.4 (0.25) | | 72.3 (70.3–74.2) | | 87.2 (85.9–88.3) | |
| **Marital status** | | | | | | | | |
| Never married | 87.3 (0.27) | <0.001 | 86.4 (0.29) | <0.001 | 37.4 (35.2–39.7) | <0.001 | 66.6 (64.4–68.8) | <0.001 |
| Married/Cohabiting | 95.2 (0.12) | | 93.5 (0.12) | | 67.5 (66.5–68.4) | | 89.2 (88.6–89.7) | |
| **Separated/Divorced/Widowed** | 95.0 (0.35) | | 93.5 (0.21) | | 67.0 (64.2–69.7) | | 87.6 (86.5–88.6) | |
| **Education level** | | | | | | | | |
| No formal schooling | 89.1 (0.70) | <0.001 | 89.9 (0.39) | <0.001 | 45.1 (38.9–51.4) | <0.001 | 76.2 (73.8–78.4) | <0.001 |
| Primary | 91.8 (0.21) | | 94.4 (0.18) | | 54.4 (52.4–56.3) | | 88.6 (87.7–89.4) | |
| Secondary | 92.7 (0.18) | | 93.0 (0.16) | | 58.1 (56.7–59.5) | | 86.5 (85.5–87.3) | |
| Higher | 95.7 (0.19) | | 91.0 (0.18) | | 69.5 (68.1–70.9) | | 84.1 (82.9–85.2) | |
| **Wealth Index** | | | | | | | | |
| Poorest | 87.4 (0.14) | <0.001 | 88.6 (0.17) | <0.001 | 36.6 (35.1–38.1) | <0.001 | 76.0 (74.9–77.2) | <0.001 |
| Poorer | 91.7 (0.21) | | 92.9 (0.19) | | 54.4 (52.6–56.2) | | 87.1 (86.0–88.2) | |
| Middle | 93.8 (0.25) | | 93.9 (0.22) | | 63.8 (61.7–65.8) | | 87.2 (86.0–88.4) | |
| Richer | 96.4 (0.25) | | 93.6 (0.25) | | 72.4 (70.3–74.4) | | 87.5 (86.0–88.8) | |
| Richest | 98.5 (0.31) | | 92.9 (0.28) | | 79.4 (77.4–81.3) | | 88.6 (87.0–90.0) | |
| **Natural regions** | | | | | | | | |
| Jungle | 90.6 (0.18) | <0.001 | 90.7 (0.16) | <0.001 | 50.4 (48.7–52.0) | <0.001 | 83.2 (82.2–84.2) | <0.001 |
| Mountain Range | 90.2 (0.16) | | 90.5 (0.16) | | 48.0 (46.5–49.5) | | 80.3 (79.3–81.2) | |
| Rest of Coast | 95.0 (0.18) | | 94.0 (0.15) | | 67.1 (65.7–68.5) | | 89.0 (88.1–89.8) | |
| Metropolitan Lima | 95.8 (0.25) | | 93.3 (0.22) | | 70.0 (68.1–71.8) | | 87.3 (86.1–88.5) | |
| **Area** | | | | | | | | |
| Rural | 88.3 (0.15) | <0.001 | 89.4 (0.17) | <0.001 | 40.0 (38.5–41.5) | <0.001 | 78.0 (76.9–79.0) | <0.001 |
| Urban | 94.9 (0.13) | | 93.2 (0.12) | | 66.7 (65.7–67.8) | | 87.3 (86.6–87.9) | |
| **Altitude (meters above sea level)** | | | | | | | | |
| 0–499 | 95.1 (0.16) | <0.001 | 93.3 (0.14) | <0.001 | 67.2 (66.0–68.4) | <0.001 | 87.6 (86.8–88.3) | <0.001 |
| 500–1499 | 92.2 (0.33) | | 92.2 (0.28) | | 56.3 (53.4–59.1) | | 85.8 (84.2–87.3) | |
| 1500–2999 | 91.4 (0.24) | | 91.7 (0.23) | | 52.3 (50.1–54.4) | | 83.4 (82.2–84.7) | |
| 3000 or more | 89.3 (0.20) | | 89.6 (0.20) | | 45.0 (43.1–46.9) | | 77.9 (76.6–79.2) | |
| **Chronic disease** | | | | | | | | |
| No | 91.9 (0.12) | <0.001 | 91.1 (0.11) | <0.001 | 55.8 (54.8–56.8) | <0.001 | 83.5 (82.8–84.2) | <0.001 |
| Yes | 98.8 (0.23) | | 97.6 (0.23) | | 78.9 (77.3–80.4) | | 93.1 (92.1–93.9) | |
| **Smoker[e]** | | | | | | | | |
| No | 93.5 (0.12) | 0.331 | 92.4 (0.10) | 0.002 | 61.7 (60.7–62.7) | 0.387 | 85.5 (84.9–86.1) | 0.906 |
| Yes | 93.8 (0.29) | | 94.2 (0.57) | | 60.6 (58.4–62.7) | | 85.7 (82.5–88.3) | |

Weight specifications included the expansion factor and the ENDES sample specifications.

SE: standard error

Men (n = 26,789)

Women (n = 35,349)

[a]P-value for ANOVA test

[b]P-value for chi2 test

[c]Between men and women

[d]By WHO Population

[e]Smoked during the previous 30 days

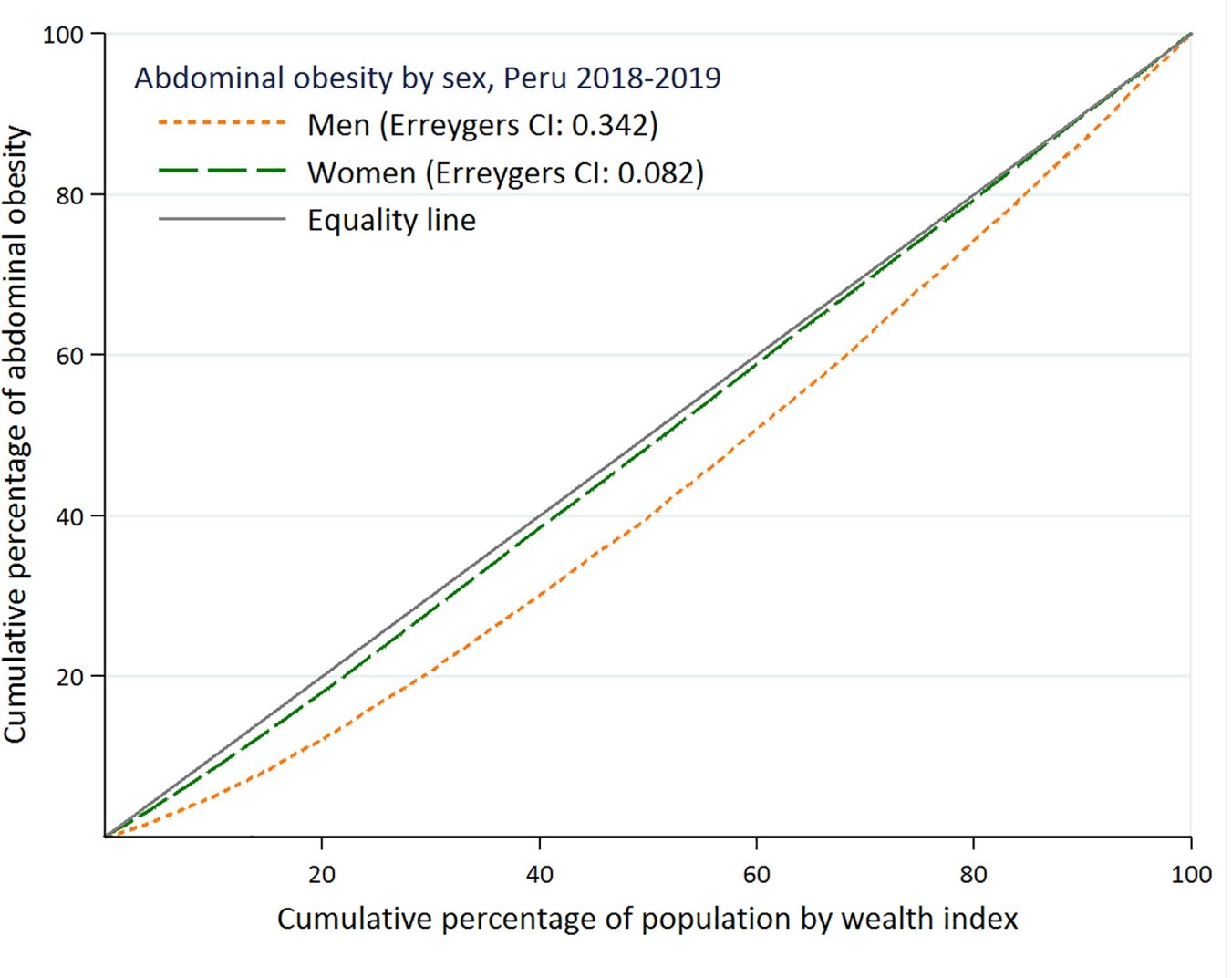

**Fig 2. Study abdominal obesity concentration curves in men and women.**

men was 0.342 (SE 0.0065) being 0.082 (SE 0.0043) in women. When we decomposed the inequality in AO, we found that the major contributors were the wealth index (men 37.2%, women 45.6%), education level (men 34.4%, women 41.4%) and living in an urban setting (men 22%, women 57.5%) (Fig 2) (Table 3).

### Prevalence of AO using different cut-off points

Using the IDF cut-off points, the overall prevalence of AO in this study was 73.8%. When using the ATP III and LASO cut-off points, the percentage of AO decreased to 43.6% and 40.6%, respectively (S1 Table). For all cut-off points, the coast region concentrated the administrative regions with the highest prevalence of AO, which were: Tumbes, Lima, Callao, Moquegua, Tacna, Arequipa, and Ica. It should be mentioned that Moquegua and Tacna, in particular, maintained a AO prevalence greater than 50% even with the more flexible cut-off points (S2 Table).

**Table 3. Decomposition of concentration indices for men and women.**

| Variable | Men | | | | Women | | | |
|---|---|---|---|---|---|---|---|---|
| | Elasticity | ECIs | Contribution | % | Elasticity | ECIs | Contribution | % |
| **Age groups, years** | | | | | | | | |
| 18–29 | Base | Base | Base | Base | Base | Base | Base | Base |
| 30–59 | 0.490 | 0.053 | 0.026 | 7.7 | 0.308 | 0.056 | 0.017 | 20.9 |
| 60 or more | 0.164 | 0.003 | 0.001 | 0.1 | 0.067 | 0.021 | 0.001 | 1.7 |
| **Marital status** | | | | | | | | |
| Never married | Base | Base | Base | Base | Base | Base | Base | Base |
| Married/Cohabiting | 0.475 | -0.012 | -0.005 | -1.6 | 0.314 | -0.078 | -0.024 | -29.8 |
| **Separated/Divorced/Widowed** | 0.050 | -0.032 | -0.002 | -0.5 | 0.087 | -0.009 | -0.001 | -1.0 |
| **Education level** | | | | | | | | |
| No formal school | Base | Base | Base | Base | Base | Base | Base | Base |
| Primary | 0.059 | -0.336 | -0.020 | -5.8 | 0.069 | -0.307 | -0.021 | -26.0 |
| Secondary | 0.183 | -0.160 | -0.029 | -8.6 | 0.115 | -0.049 | -0.006 | -6.8 |
| Higher | 0.221 | 0.533 | 0.118 | 34.4 | 0.067 | 0.506 | 0.034 | 41.4 |
| **Wealth Index** | | | | | | | | |
| Poorest | Base | Base | Base | Base | Base | Base | Base | Base |
| Poorer | 0.090 | -0.354 | -0.032 | -9.3 | 0.055 | -0.349 | -0.019 | -23.4 |
| Middle | 0.139 | 0.003 | 0.000 | 0.1 | 0.048 | -0.014 | -0.001 | -0.8 |
| Richer | 0.177 | 0.335 | 0.059 | 17.4 | 0.049 | 0.312 | 0.015 | 18.7 |
| Richest | 0.205 | 0.621 | 0.127 | 37.2 | 0.058 | 0.651 | 0.037 | 45.6 |
| **Natural regions** | | | | | | | | |
| Jungle | Base | Base | Base | Base | Base | Base | Base | Base |
| Mountain Range | -0.008 | -0.320 | 0.003 | 0.8 | 0.020 | -0.362 | -0.007 | -8.7 |
| Rest of Coast | 0.058 | 0.040 | 0.002 | 0.7 | 0.040 | 0.044 | 0.002 | 2.1 |
| Metropolitan Lima | 0.039 | 0.496 | 0.020 | 5.7 | 0.019 | 0.518 | 0.010 | 12.1 |
| **Area** | | | | | | | | |
| Rural | Base | Base | Base | Base | Base | Base | Base | Base |
| Urban | 0.134 | 0.562 | 0.075 | 22.0 | 0.086 | 0.552 | 0.047 | 57.5 |
| **Altitude (meters above sea level)** | | | | | | | | |
| 0–499 | Base | Base | Base | Base | Base | Base | Base | Base |
| 500–1499 | -0.002 | -0.072 | 0.000 | 0.0 | 0.000 | -0.061 | 0.000 | 0.0 |
| 1500–2999 | -0.008 | -0.099 | 0.001 | 0.2 | -0.006 | -0.098 | 0.001 | 0.7 |
| 3000 or more | -0.026 | -0.238 | 0.006 | 1.8 | -0.028 | -0.281 | 0.008 | 9.7 |
| **Chronic disease** | | | | | | | | |
| No | Base | Base | Base | Base | Base | Base | Base | Base |
| Yes | 0.145 | 0.087 | 0.013 | 3.7 | 0.062 | 0.050 | 0.003 | 3.7 |
| **Smoker[a]** | | | | | | | | |
| No | Base | Base | Base | Base | Base | Base | Base | Base |
| Yes | 0.011 | 0.029 | 0.000 | 0.1 | 0.003 | 0.061 | 0.000 | 0.2 |
| **Residual** | | | -0.021 | | | | -0.280 | |

Weight specifications included the expansion factor and the ENDES sample specifications.

[a]Having smoked during the previous 30 days

## Discussion

Our study examines the socioeconomic inequalities in AO and their determinants among Peruvian adults. We found that AO is more concentrated among the wealthiest individuals,

with higher inequality being found in men. The major contributors of inequality were the wealth index, higher education and living in an urban setting.

We found that the wealth index was the main driver of inequality in AO. This finding is consistent with studies in other upper-middle-income countries such as Indonesia [18] and China [36]. In addition, we found a higher prevalence of AO in coastal cities. This result may be due to the fact that coastal cities are the wealthiest compared to jungle and mountain regions. The incidence of monetary poverty in the coast, jungle and mountain are 13.8%, 25.8% and 29.3% respectively [37]. The annual GDP per capita in 2017 for the coast, the mountain and the jungle were 9,764, 3,780 and 2,442 US dollars [38]. The prevalence of AO in coastal cities may be higher because people with a high level of well-being, especially those in an urban environment, are more likely to engage in unhealthy routine behaviors such as sedentarism and consumption of high caloric food [39], in addition, these people are exposed to a great amount of advertising of caloric dense food [40]. Furthermore, the wealthiest individuals tend to suffer 'technological sedentarism' due to greater access to their own form of motorized transport and greater quantities of office work which promote less physical activity [41]. In short, our country faces a series of economic and nutritional changes that promote an increase of AO, which seems to be more pronounced in the coast.

We found a higher prevalence of AO among more educated individuals. This coincides with other studies including a systematic review of 91 countries including Peru, in which people with a higher education in less developed countries tended to be more obese [42, 43], while in more developed countries such as China the opposite is true [44]. A Peruvian study found that the population with a higher education level was 1.5 times less likely to perform physical activity [45], which could be due to the close association with office jobs, in which individuals remain seated for long hours. Indeed, a study in France reported that approximately 7.5 hours a day and 37.5 hours a week are spent sitting in front of a computer [46]. Other factors such as a greater workload and extended hours could limit access to a more balanced diet prepared at home, leading to greater demand for high-calorie fast food [47], favoring the development of obesity. Therefore, it is important to promote the adoption of healthy lifestyles across all educational levels, providing information about AO and its consequences.

Another relevant finding was that residents of both sexes in urban areas had higher AO. In Perú, the population in urban areas is 1.9 times more likely to have lower physical activity when compared to that living in rural areas [45]. In the United States it has been reported that occupational physical activity in metropolitan areas has been reduced by about 120 kcal/day [48]. Furthermore, the supply of vehicles in the capital of Peru is 175.8 vehicles per 1000 inhabitants, while in the jungle and mountains, such as Loreto and Puno, the rate only reaches 5.24 and 33.37 vehicles per 1000 inhabitants, respectively [49]. Roads in rural areas are usually very rough with insufficient access ways, and residents of this area are thereby conditioned to do more physical activity as part of their daily commute [50]. Access to communication technologies and the consequent time spent in front of a screen could be another contributor to this observation: 55.9% of homes in Metropolitan Lima have cable TV while in rural areas this is available in only 11.7% of homes. Likewise, only 4.6% of rural homes have access to internet while in Metropolitan Lima it reaches 58.7% [51]. Gaps in access to these technologies can therefore favor sedentarism in urban areas.

Finally, we observed a higher prevalence of AO in women, which is consistent with other studies [18, 36, 52]. The prevalence of AO was greatest among women belonging to the 30–59 age group, which might be explained by the decline in regular exercise and the use of diets over time [18]. Of note was the large proportion of women aged 18–29 years of age with AO compared to their male counterparts. In Peru, the prevalence of fertility in women between 20–34 years of age exceeds 60% in both rural and urban areas [53], and this can cause

progressive weight gain associated with pregnancy. In Latin America, the participation of women in the labor market is still lagging compared to other Organization for Economic Cooperation and Development (OECD) countries [54]. Even among individuals with higher education it is estimated that 86.2% of men in Perú are employed vs. 74.5% of women. This percentage drops to 67.8% among women with a primary education but remains around 82.2% for their male counterparts. Likewise, the average monthly salary of men of 1846 PEN (487.69 USD) is higher than the 1322 PEN (349.26 USD) earned by women [55]. Therefore, Peruvian women may be less motivated to pursue their professional career path due to inequalities in the workforce and a lack of monetary incentive; in particular women with a primary education would most likely be housewives or unemployed, thereby being less physically active [56]. The persistence of traditional gender roles, in which the male is usually engaged in physically demanding work [57], contributes to women being more obese in rural areas, being widely accepted among women because of its association with pregnancy and child-bearing [58]. Due to social beliefs in countries such as Indonesia and Perú [59], obesity might be seen as a symbol of good economic status [18]. All this could contribute to women more frequently presenting AO, and thus, it is important to develop interventions aimed at these gender disparities, considering the social and economic context of Peru.

Among the limitations of this study, due to it being a cross sectional, secondary analysis, we were unable to establish causal relationships. However, ENDES is the most representative national demographic health survey, providing updated information on the country's demographic situation. The second limitation is that there are no established cut-off values for determining AO using waist circumference for the Peruvian population. This may bias the results because the definitions used for estimations are not adapted to the population and context in Peru. In an attempt to solve this, we estimated the prevalence of AO with different cut-off points such as those of ATP III and LASO. Despite the potential limitations, the results have important implications for equity and health policy.

## Conclusion

The socioeconomic characteristics of the population impact the distribution of AO in Peru. This inequality is more prominent among men although AO is more prevalent among women. A higher wealth index, higher levels of education and living in urban areas were the main contributors to AO inequality. Peru is going through an economic and nutritional transition that is generating a change in the development and distribution of AO. Understanding this problem could serve as the basis for the creation of health programs focused on reducing inequality gaps among the population at greatest risk.

## Supporting information

**S1 Table. Prevalence of abdominal obesity according to socioeconomic characteristics and based on the Third Adult Treatment Panel and the Latin American Consortium of Studies in Obesity cut-off points.**
(DOCX)

**S2 Table. Prevalence of abdominal obesity in men and women according to the administrative region of residence (ENDES 2018–2019) and based on IDF, the guidelines of the Third Adult Treatment Panel (ATP III) and the Latin American Consortium of Studies in Obesity (LASO).**
(DOCX)

## Acknowledgments

We thank Donna Pringle for reviewing the language and style of the manuscript.

## Author Contributions

**Conceptualization:** Akram Hernández-Vásquez.

**Data curation:** Akram Hernández-Vásquez.

**Formal analysis:** Akram Hernández-Vásquez, Carlos Rojas-Roque.

**Methodology:** Akram Hernández-Vásquez, Carlos Rojas-Roque.

**Project administration:** Akram Hernández-Vásquez.

**Validation:** Marioli Y. Farro-Maldonado, Glenda Gutiérrez-Pérez, Antonio Barrenechea-Pulache, Marilina Santero, Carlos Rojas-Roque, Diego Azañedo.

**Visualization:** Marioli Y. Farro-Maldonado, Glenda Gutiérrez-Pérez, Akram Hernández-Vásquez, Antonio Barrenechea-Pulache, Marilina Santero, Carlos Rojas-Roque, Diego Azañedo.

**Writing – original draft:** Marioli Y. Farro-Maldonado, Glenda Gutiérrez-Pérez, Antonio Barrenechea-Pulache, Marilina Santero, Carlos Rojas-Roque, Diego Azañedo.

**Writing – review & editing:** Marioli Y. Farro-Maldonado, Glenda Gutiérrez-Pérez, Akram Hernández-Vásquez, Antonio Barrenechea-Pulache, Marilina Santero, Carlos Rojas-Roque, Diego Azañedo.

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
