## [Decision Letter · Decision Letter 0]

9 Mar 2021

PONE-D-20-35836

Socioeconomic inequalities in abdominal obesity among Peruvian adults

PLOS ONE

Dear Dr. Barrenechea-Pulache,

Thank you for submitting your manuscript to PLOS ONE. After careful consideration, we feel that it has merit but does not fully meet PLOS ONE’s publication criteria as it currently stands. Therefore, we invite you to submit a revised version of the manuscript that addresses the points raised during the review process.

We look forward to receiving your revised manuscript.

Kind regards,

Isil Ergin, Assoc. Prof.

Academic Editor

PLOS ONE

Additional Editor Comments:

1. The use of “influence” or similar terminologies of effect should be revaluated as this is a cross sectional study design.

2. Rewording of the aim should be considered as it does not include an evaluation of the cut offs.

3. Regarding the use of Concentration Index to evaluate the inequalities, please define what advantages and disadvantages it has for such evaluation.

4. The additions recommended for a clearer view on sampling, data presentation and statistical analysis should be dealt in detail.

5. Language editing needed.

Journal Requirements:

3. We note that Figure S2 in your submission contain map images which may be copyrighted. All PLOS content is published under the Creative Commons Attribution License (CC BY 4.0), which means that the manuscript, images, and Supporting Information files will be freely available online, and any third party is permitted to access, download, copy, distribute, and use these materials in any way, even commercially, with proper attribution. For these reasons, we cannot publish previously copyrighted maps or satellite images created using proprietary data, such as Google software (Google Maps, Street View, and Earth). For more information, see our copyright guidelines: http://journals.plos.org/plosone/s/licenses-and-copyright.

3.1.    You may seek permission from the original copyright holder of Figure S2 to publish the content specifically under the CC BY 4.0 license. 

3.2.    If you are unable to obtain permission from the original copyright holder to publish these figures under the CC BY 4.0 license or if the copyright holder’s requirements are incompatible with the CC BY 4.0 license, please either i) remove the figure or ii) supply a replacement figure that complies with the CC BY 4.0 license. Please check copyright information on all replacement figures and update the figure caption with source information. If applicable, please specify in the figure caption text when a figure is similar but not identical to the original image and is therefore for illustrative purposes only.

Reviewers' comments:

Reviewer's Responses to Questions

**Comments to the Author**

1. Is the manuscript technically sound, and do the data support the conclusions?

Reviewer #1: Partly

Reviewer #2: Yes

2. Has the statistical analysis been performed appropriately and rigorously? 

Reviewer #1: No

Reviewer #2: Yes

3. Have the authors made all data underlying the findings in their manuscript fully available?

Reviewer #1: Yes

Reviewer #2: Yes

4. Is the manuscript presented in an intelligible fashion and written in standard English?

Reviewer #1: Yes

Reviewer #2: No

5. Review Comments to the Author

Reviewer #1: The statistical analysis is based primarily on the Lorenze curve and the ECI with appropriate decomposition. The authors have concluded that, of the socioeconomic characteristics of the population, the variables which impact the inequality of the distribution of AO in Peru are the wealth index, education level and area of residence. The major comparison seems to be between the sexes. This appears in a univariate format for all the results.

There are several variables involved. However, the analysis lacks sophistication. There may be some age adjustments involved. However, there are no obvious multivariate models presented or the investigators have failed to adequately explain any multivariate or confounding relationships if they exist. This certainly is a major limitation of this cross sectional presentation. The manuscript should be re-assessed statistically.

Reviewer #2: Thank you for having me reviewing the manuscript. I have uploaded my review as an attachment. I hope I will be of assistance to the authors and I think the paper could be improved with attention to some of these matters.

6. PLOS authors have the option to publish the peer review history of their article (what does this mean?). If published, this will include your full peer review and any attached files.

Reviewer #1: No

Reviewer #2: No

---

## [Author Response · Author response to Decision Letter 0]

11 May 2021

May 10th, 2021

Dear Isil Ergin, Assoc. Prof.

Academic Editor

PLOS ONE

Ref: Submission [PONE-D-20-35836]

Title: "Socioeconomic inequalities in abdominal obesity among Peruvian adults."

We thank the reviewers and the Editor-in-Chief for their helpful comments and suggestions provided for our manuscript. All comments and suggestions have been addressed in the new revised version of the manuscript. All the comments and changes are described below according to the revision process. 

Journal requirements

1. Please ensure that your manuscript meets PLOS ONE's style requirements, including those for file naming. The PLOS ONE style templates can be found at: https://journals.plos.org/plosone/s/file?id=wjVg/PLOSOne_formatting_sample_main_body.pdf

Answer: Thank you for bringing this to our attention. We have reviewed the journal style requirements and have changed the formatting of the tables, the reference style and file naming accordingly.

Answer: Thank you for bringing this to our attention. We have added a data sharing statement section in materials and methods. Likewise, we have updated the data availability statement in our cover letter modeling it after what is published in other articles found in PLos One that used the ENDES database [1,2]. 

1. Accinelli RA, Leon-Abarca JA. Age and altitude of residence determine anemia prevalence in Peruvian 6 to 35 months old children. PLoS One [Internet]. 2020 Jan 1 [cited 2021 May 5];15(1):e0226846. Available from: https://doi.org/10.1371/journal.pone.0226846

2. Chambergo-Michilot D, Rebatta-Acuña A, Delgado-Flores CJ, Toro-Huamanchumo CJ. Socioeconomic determinants of hypertension and prehypertension in Peru: Evidence from the peruvian demographic and health survey. PLoS One [Internet]. 2021 Jan 1 [cited 2021 May 5];16(1 January):e0245730. Available from: https://doi.org/10.1371/journal.pone.0245730

Data sharing statement:

“The database used in this study is open access and available on the INEI website at http://iinei.inei.gob.pe/microdatos/.”

In our cover letter it now reads:

“Data Availability statement 

The database is freely accessible from the National Institute of Statistics and Informatics (INEI) website (http://iinei.inei.gob.pe/microdatos/). The information can be obtained by entering the survey query tab and selecting the ENDES 2018-2019; data is obtained from modules #64, #65 and #414.”

3. We note that Figure S2 in your submission contains map images which may be copyrighted. All PLOS content is published under the Creative Commons Attribution License (CC BY 4.0), which means that the manuscript, images, and Supporting Information files will be freely available online, and any third party is permitted to access, download, copy, distribute, and use these materials in any way, even commercially, with proper attribution. For these reasons, we cannot publish previously copyrighted maps or satellite images created using proprietary data, such as Google software (Google Maps, Street View, and Earth). For more information, see our copyright guidelines: http://journals.plos.org/plosone/s/licenses-and-copyright.

3.1. You may seek permission from the original copyright holder of Figure S2 to publish the content specifically under the CC BY 4.0 license.

3.2. If you are unable to obtain permission from the original copyright holder to publish these figures under the CC BY 4.0 license or if the copyright holder’s requirements are incompatible with the CC BY 4.0 license, please either i) remove the figure or ii) supply a replacement figure that complies with the CC BY 4.0 license. Please check copyright information on all replacement figures and update the figure caption with source information. If applicable, please specify in the figure caption text when a figure is similar but not identical to the original image and is therefore for illustrative purposes only.

Answer: Thank you for bringing this to our attention. We have decided to replace the figures for an additional table (to be considered as supplementary material) in order to avoid any potential copyright infringements and provide more specific information about each sub-national administrative unit. This table specifies the prevalence of abdominal obesity in men and women according to the region of residence based on IDF, ATP III and LASO cut-off points.

Changes have been made to the final paragraph of the results section, incorporating a general interpretation of the results of this new table.

It now reads: “Using the IDF cut-off points, the overall prevalence of AO in this study was 73.8%. When using the ATP III and LASO cut-off points, the percentage of AO decreased to 43.6% and 40.6%, respectively (S1 Table). For all cut-off points, the coast region concentrated the administrative regions with the highest prevalence of AO, which were: Tumbes, Lima, Callao, Moquegua, Tacna, Arequipa, and Ica. It should be mentioned that Moquegua and Tacna, in particular, maintained an AO prevalence greater than 50% even with the more flexible cut-off points (S2 Table).”

Comments and suggestions made by the Editor

1. The use of “influence” or similar terminologies of effect should be reevaluated as this is a cross sectional study design.

Answer: Thank you for your comment. We agree with the reviewer that since our study design is cross sectional we cannot establish causal relationships. We have modified the sentence in materials and methods. 

The second paragraph of study population and design now reads:

“..We used information compiled from both the household and the health questionnaires to carry out a secondary analysis to determine the prevalence and inequalities in the distribution of AO in adults. ...”

2. Rewording of the aim should be considered as it does not include an evaluation of the cut offs.

Answer: We agree with your comment and have included the cut-off points in the Abstract and Introduction. 

The abstract now reads “...Thus, our aim was to analyze the socioeconomic inequalities in AO distribution defined using the International Diabetes Federation (IDF) cut-off points in Peruvian adults in 2018-2019.” 

The last paragraph of the introduction now reads: “Therefore, this article aimed to analyze the socioeconomic inequalities in AO distribution, using the International Diabetes Federation (IDF) cut-off points for South and Central America [7] in Peruvian adults using information from the 2018-2019 ENDES. …”

3. Regarding the use of Concentration Index to evaluate the inequalities, please define what advantages and disadvantages it has for such evaluation.

Answer: Thanks for this comment. We have added further details about the disadvantages and advantages of using the concentration index In the methods section. 

The following text was added to the second paragraph of the Statistical Analysis subsection:

“One shortfall of the CI is that in the scenario of data contamination, the index is sensitive to extreme values at one or both tails of the distribution [29]. However, its main advantages are that it reflects the socioeconomic dimension to inequalities in health and the experiences of the entire population. The CI is sensitive to changes in the distribution of the population across socioeconomic groups, and, therefore, has been widely used to measure inequality within health economics.”

We have added a reference:

29. Giorgi GM, Gigliarano C. The Gini concentration index: A review of the inference literature: The Gini concentration index. J Econ Surv. 2017;31(4):1130–1148.

4. The additions recommended for a clearer view on sampling, data presentation and statistical analysis should be dealt in detail.

Answer. We appreciate this comment. We have carefully reviewed the details regarding sampling, data presentation and statistical analysis and have modified the corresponding section to make it more specific. We have added a section describing the population characteristics, inclusion and exclusion criteria and a flowchart of the study population included.

5. Language editing needed.

Answer: We thank the editor for bringing this to our attention, we have edited the text with the aid of a native speaker so as to solve any language errors. This has been referenced in the acknowledgments section.

Comments and suggestions made by the Reviewer 1

1. The statistical analysis is based primarily on the Lorenze curve and the ECI with appropriate decomposition. The authors have concluded that, of the socioeconomic characteristics of the population, the variables which impact the inequality of the distribution of AO in Peru are the wealth index, education level and area of residence. The major comparison seems to be between the sexes. This appears in a univariate format for all the results. There are several variables involved. However, the analysis lacks sophistication. There may be some age adjustments involved. However, there are no obvious multivariate models presented or the investigators have failed to adequately explain any multivariate or confounding relationships if they exist. This certainly is a major limitation of this cross sectional presentation. The manuscript should be re-assessed statistically.

Answer: We thank the reviewer for this comment. As we noted, in the manuscript the results of Table 3 appear as a univariate analysis and in the methods sections there is no evidence that a multivariable analysis was performed .However, by checking our analysis and the codes developed for the decomposition, we confirm that our decomposition analysis was done using the generalized linear models (GLM) approach. In comparison to other approaches such as the probit estimations or the ordinary least squares, GLM has shown to be the best choice when decomposing inequalities using a binary outcome [1]. In addition, the decomposition analysis was adjusted for the socioeconomic (wealth index) and demographic variables (age, marital status, educational level (no formal school/primary/secondary/higher), presence of chronic disease (yes/no), smoker (yes/no), area of residence (urban/rural), altitude above sea level of the housing conglomerate, natural region (jungle/mountain range/rest of coast/Metropolitan Lima) included in the analysis. To state the methods of the decomposition analysis more clearly, we have expanded our description and detailed the econometric equation estimated. All the methodology was based on the book of O’Donnell and colleagues [2]. The book deals with analyzing health equity using household survey data. 

References

1. Yiengprugsawan V, Lim LL, Carmichael GA, Dear KB, Sleigh AC. Decomposing socioeconomic inequality for binary health outcomes: an improved estimation that does not vary by choice of reference group. BMC Res Notes (2010) 3:57. doi:10.1186/1756-0500-3-57.

2. O’Donnell O, van Doorslaer E, Wagstaff A, Lindelow M. Analyzing Health Equity Using Household Survey Data: A Guide to Techniques and Their Implementation. The World Bank, Washington, D.C; 2008. 234p

In the subsection Statistical Analysis of the manuscript, we have added the following:

“The decomposition was performed based on generalized linear models (GLM). In comparison to other approaches such as probit estimations or the ordinary least squares, GLM has shown to be the best choice when decomposing inequalities using a binary outcome [33]. This study decomposes the inequality of abdominal obesity using the following equation:

Where ECI is the Erreygers concentration index, XX is the mean of the explanatory variables included in the decomposition (the socioeconomic and demographic factors), XX is the partial effect evaluated at the sample means, XX is the mean of the concentration index, and XX is the generalized concentration index of the stochastic term of error. Equation [c] reflects that an explanatory variable contributes to the inequality in AO when this variable is correlated with AO and is not equally distributed across the wealth index. The contribution of the explanatory variable to the inequality in AO depends on the absolute value of the partial effect and the unequal distribution of the explanatory variable with respect to household income per-capita. A positive sign of the partial effect means that the explanatory variable contributes to an increase in the inequality observed, and vice versa [27]. All analyses were performed using Stata version 14.2.”

One reference has been added:

33. Yiengprugsawan V, Lim LL, Carmichael GA, Dear KB, Sleigh AC. Decomposing socioeconomic inequality for binary health outcomes: an improved estimation that does not vary by choice of reference group. BMC Res Notes. 2010;3(1):57.

Comments and suggestions made by the Reviewer 2

Overall comments.

1. For every study cited in this manuscript, I suggest adding detailed information on where the study was conducted and/or the study population (i.e., age, sex, etc.).

Answer. Thank you for your suggestion. We have added information regarding the studies cited, whenever we deem it pertinent to be specific. Changes have been made throughout the introduction and discussion sections. Specifically, information has been added regarding the population under study (age and country of origin if information regarding the latter had not been mentioned previously). Adding sex details was only considered if any of the cited studies only included men or women exclusively.

2. Many sentences (especially in the discussion section) that based on the reference was not explained properly. Please add the detail on the reference studies in the sentence to help readers understand.

Answer. Thank you for pointing this out. We have revisited the introduction and discussion sections to add details in the sentences that are based on the referenced papers. Likewise, we have added further information to the discussion section. 

3. I noticed some long sentences (e.g., Line 270-274) which are difficult to follow. Please re-phrase the long sentences and make it short and clear.

Answer. Thank you for the observation. We have rephrased this sentence.

It now reads:

“... A Peruvian study found that the population with a higher education level was 1.5 times less likely to perform physical activity [45], which could be due to the close association with office jobs, in which individuals remain seated for long hours. Indeed, a study in France reported that approximately 7.5 hours a day and 37.5 hours a week are spent sitting in front of a computer [46].”

Detailed comments.

1. The sentence in the abstracts section should be revised to make it short but clear, especially the methods part.

Answer. Thank you for the suggestion, we have shortened and rewritten the abstract sections for better understanding.

The methods section of the abstract now reads:

“This was a cross-sectional study using data from the 2018-2019 Demographic and Family Health Survey (ENDES) of Peru. We analyzed a representative sample of 62,138 adults over 18 years of age of both sexes from urban and rural areas. Subjects were grouped into quintiles of the wealth to calculate a concentration curve and the Erreygers Concentration Index (ECI) in order to measure the inequality of AO distribution. Finally, we performed a decomposition analysis to evaluate the major determinants of inequalities.”

2. Line 26, 38. AO and ECI was used first here, but no information what AO and ECI abbreviation stands for in the abstracts section.

Answer: We thank the reviewer for bringing this to our attention. We added the full name in the cited sections.

3. The ‘Methods’ part in abstracts need detailed information on where the study conducted, who is the study subjects and their age range.

Answer: Thank you for your comment. As mentioned above we have rewritten the abstract and detailed the country of study (Peru) and the age of the population studied (over 18 years old). 

4. Line 30-31: “We analysed a representative sample of adults residing in urban/rural households (31 553 and 30 585 from the 2018 and 2019 dataset respectively).”

a. Adults age? Men and women?

b. Urban/rural household in Peru? Is it urban AND rural? Or is it urban OR rural?

c. 31 553 and 30 585 is the total number of individuals or household? Perhaps you should add a comma in all the number to make it easier to read (31,553 and 30,585)

Answer: We agree with your suggestion and have modified this section to better characterize our study population.

It now reads: “This was a cross-sectional study using data from the 2018-2019 Demographic and Family Health Survey (ENDES) of Peru. We analyzed a representative sample of 62,138 adults over 18 years of age of both sexes from urban and rural areas.”

5. Line 43. The conclusion should be clearer. E.g., Abdominal obesity in Peru was more prevalent among women. Socioeconomic inequality in abdominal obesity exists among Peruvian favouring the advantaged group. The inequality gap is less prominent among women, showing obesity being more common among the poor. Etc.

Answer: Thank you for the observation. We have modified the conclusion section of the abstract to make it more specific.

It now reads: “In Peru the wealthy concentrate a greater percentage of AO. The inequality gap is greater among men, yet AO is more prevalent among women. The variables that most contribute to inequality were the wealth index, education level and area of residence. There is a need for effective individual and community interventions to reduce these inequalities.”

6. Line 59: Which developing countries does the author refer to? Need a Reference to support this sentence.

Answer: Thank you for bringing this to our attention, we have modified the cited sentence and added a reference to clarify the message. 

It now reads: “This disease threatens to overload the economic and resolutive capacity of health systems, particularly in Latin American countries in which budgets assigned to health are very limited, being of around 7.9% of the Gross Domestic Product (GDP) in 2018 compared to 16.9% in the United States [8].”

We have added 1 reference:

8. The World Bank. Current health expenditure (% of GDP) [Internet]. 2018 [cited 2021 May 9]. Available from: https://data.worldbank.org/indicator/SH.XPD.CHEX.GD.ZS

7. I suggest adding a paragraph on information about the study settings (in the introduction or methods section), i.e., Total population, number of cities in Peru, the economy classification by The World Bank, the population health, etc. This brief background information of Peru will increase the reader’s understanding of the study subject and the contexts.

Answer. We appreciate the reviewer's comment. We added a paragraph in the beginning of the study population and design section to better characterize Peru and its population.

It now reads: “Peru is a country divided into 24 sub-national administrative units, known as “administrative regions” and 1 constitutional province. The territorial area is 1,285,215.60 km2 and borders Ecuador, Colombia, Brazil, Bolivia and Chile. The total population in 2019 was 32,131,400 million people, being the 7th most populated country in Latin America [19]. Peru can be divided into three natural regions: the coast, which concentrates 58% of the national population and many of the most developed cities including Lima, the capital [20]; the jungle, which is difficult to access due to the rugged terrain of the Amazon and whose population has insufficient access to basic services; and the highlands, the Andean area which presents the highest level of monetary poverty in the country. According to The World Bank the economy of Peru belongs to the upper middle income (gross national income per capita between $4,046 and $12,535) [21]. In 2018, 5.2% of the GDP was invested in health, being one of the lowest compared to other South American countries such as Colombia, Chile and Brazil [8].” 

Notes. 3 references were added in the manuscript: 

19. Instituto Nacional de Estadística e Informática.[State of the Peruvian population. 2019]. Peru: National Institute of Statistics and Informatics. Instituto Nacional de Estadística e Informática. 2019. Available from: https://www.inei.gob.pe/media/MenuRecursivo/publicaciones_digitales/Est/Lib1671/libro.pdf

20. Instituto Nacional de Estadística e Informática. [Peru: Final Results of the 2017 National Census]. Instituto Nacional de Estadística e Informática. 2018. Available from: https://www.inei.gob.pe/media/MenuRecursivo/publicaciones_digitales/Est/Lib1544/

21. The World Bank. World Bank Country and Lending Groups. The World Bank. 2021. Available from: https://datahelpdesk.worldbank.org/knowledgebase/articles/906519-world-bank-country-and-lending-groups

8. Line 104. ‘36 760’ is the number of individuals or household?

Answer. 

We appreciate the reviewer's comment, the ENDES utilizes a two step sample, the second step of which is at the household level from here at least 1 individual 15 years and older is included for the survey. We have modified the second paragraph of the study population and design to better reflect this. 

It now reads: “... It uses a two-stage, balanced, stratified and probabilistic sample, which is representative at national, administrative region and natural region levels. Each year studied had a sample size of 36,760 households, of which one individual 15 years of age or older was included in the survey. We used information compiled from both the household and the health questionnaires to carry out a secondary analysis to determine the prevalence and inequalities in the distribution of AO in adults. ...”

9. Line 105-106. Would the authors describe what is the inclusion and exclusion criteria? How many data extracted in the beginning? How many individuals are excluded?

Answer: We thank the reviewer for bringing this to our attention. Following the STROBE guidelines we have added a population characteristics subsection at the beginning of results and a simple flowchart to aid the reader in understanding our study sample.

It now reads:

“After applying inclusion (adults over 18 years with waist circumference measurement) and exclusion criteria (individuals <18 years, pregnant, incomplete data), a total of 31,553 and 30,585 individuals were included from the 2018 and 2019 datasets, respectively (Fig 1). We included a total of 26, 789 men and 35, 349 women in the analysis.”

10. Line 103-107. The explanation of the sampling procedure was very short. Please add a simple flowchart on how the study population generated from ENDES datasets.

Answer. Thank you for your observations. We have added a flowchart in the results section to better explain the process of how we selected our study population. 

11. Line 140-141. “To measure the socioeconomic inequality in the distribution of AO across the population grouped in wealth quintiles,”

Please rephrase the sentence here to make it easier to understand.

Answer. Thank you for bringing this to our attention. The sentence has been modified for better understanding. 

Now it reads: “To measure the socioeconomic inequality in the distribution of AO across the population, subjects were grouped into quintiles of wealth to calculate the concentration curve and the concentration index (CI).”

12. Table 1, 2 and 3. Typo in Age groups: ‘60 o more’.

I think it is better to maintain the same number of digits across all tables, whether it is 2 digits or 3 digits.

Answer. Thank you for bringing this to our attention. We have reviewed the instructions for authors and there is no specification as to the number of digits required for tables. Regardless, we have reviewed previous publications on PLoS One and found that authors usually use 1 digit for percentages as can be referenced below:

1. Accinelli RA, Leon-Abarca JA. Age and altitude of residence determine anemia prevalence in Peruvian 6 to 35 months old children. PLoS One [Internet]. 2020 Jan 1 [cited 2021 May 5];15(1):e0226846. Available from: https://doi.org/10.1371/journal.pone.0226846

2. Id DJB, Id CPN, Mcgloughlin S, Pilcher D, Sarode V V, Gatward JJ. Preparation for airway management in Australia and New Zealand ICUs during the COVID -19 pandemic. 2021;1–10. Available from: http://dx.doi.org/10.1371/journal.pone.0251523

13. Table 1. No information on smoking in table 1, however, it is mentioned in the footnote (Line 193).

Answer. Thanks for the observation. We apologize for this. There was an error at the time of entry of the tables, due to it being inserted as an excel sheet. We have substituted these for conventional tables integrated in the MS word application. All the variables included are now clearly visible.

14. Line 194. I suggest dividing the section ‘Mean waist circumference and prevalence of AO’ into two paragraphs. The first paragraph talked about mean waist circumference and the second paragraph about AO prevalence.

Answer. Thank for your observation. We separated the text into two separate paragraphs as suggested. 

The first paragraph now reads: “ The mean waist circumference was higher among men than women (93.5 cm vs. 92.3 cm, respectively, p < 0.001). [...] In the case of chronic disease and altitude in both sexes, those that reported having a chronic disease and who lived between 0-499 m.a.s.l. had the highest mean waist circumference (Table 2).”

The second paragraph now reads: “Regarding the prevalence of AO, women presented a higher prevalence than men both in general (85.1% vs. 61.1%, p < 0.001) and across all other independent variables. For women the highest prevalence of AO was concentrated among those from 30-59 years of age [...] We observed a lower prevalence of AO in men without chronic diseases compared to those who had comorbidities (55.8% vs. 78.9%, p < 0.001), while in women the rate of AO was high in both groups (83.5% vs. 93.1%, p < 0.001) (Table 2). 

15. Line 195-199. “In both groups, the mean waist circumference was higher in wealthy, older individuals, those who were separated, divorced or widowed, had a higher education, those living in lower altitudes, and subjects who self-reported chronic disease (p < 0.001).

Please re-check this sentence to match the actual value in Table 1. I can see that some information in this sentence is incorrect.

Answer: Thank you for bringing this to our attention, we believe the reviewer refers to table 2 which states the mean waist circumference found among the participants in our study. We have modified the paragraph in order to more accurately reflect our results.

It now reads: “Among men the mean waist circumference increased linearly across the categories of variables so that those with the highest wealth index, aged 60 or more, and with a higher education had the largest waist circumference. Meanwhile, among women the mean waist circumference tended to be higher among the middle categories; for example, women with a middle wealth index, those aged 30-59, and women with primary education had the highest waist circumference. In the case of chronic disease and altitude, in both sexes, individuals that reported having a chronic disease and who lived between 0-499 m.a.s.l. had the highest mean waist circumference (Table 2).”

16. Line 199-201. “The prevalence of AO among women had small variations across all the independent variables, and prevalence was consistently higher than men (85.1% vs. 61.1%, p < 0.001).”

Perhaps the author should re-phrase this sentence so it will be easier to understand by the reader.

Answer. We agree with your observation and have modified the sentence to state: “Regarding the prevalence of AO, women presented a higher prevalence than men both in general (85.1% vs. 61.1%, p < 0.001) and across all other independent variables. ”

17. Line 203-204. “By educational level, the prevalence of AO in men was positively linked with the level of education,”

What does the author meant by ‘positively linked’ in this sentence?

Answer. Thank you for bringing this to our attention. We refer to the fact that as education level increases so does the prevalence of AO. We have changed the sentence to make our point clearer: “According to educational level, the prevalence of AO in men increased linearly with the level of education ... ”

18. Line 240. “According to the IDF, at a national level, the prevalence of AO was 73.8%. When using the ATP III and LASO cut-off points, the percentage of AO decreased to 43.6% and 40.6% respectively (S1 Table).”

This sentence needs additional information e.g.: “According to the IDF cut-off points, the overall prevalence of AO in this study was 73.8%. When using the ATP III and LASO cut-off points, the percentage of AO decreased to 43.6% and 40.6% respectively (S1 Table)”

Answer. We appreciate the reviewer's comment and have added additional information to make the text clearer using your suggestion: “Using the IDF cut-off points, the overall prevalence of AO in this study was 73.8%...”

19. Line 250. “Having a greater wealth index, a higher education and living in an urban setting were the major independent determinants of inequality and were positively associated with its prevalence.”

What does the author meant by ‘were positively associated with its prevalence’? Was it meant these variables (greater wealth index, a higher education and living in an urban setting) positively associated with abdominal obesity prevalence? I do not think this manuscript studied the association between the variables and abdominal obesity prevalence.

Answer. Thanks for this comment. We agree with the reviewer that measuring the association between the mentioned variables and abdominal obesity was not part of our objective. With this in mind, we have reviewed the whole manuscript and have rewritten the indicated sentence and others with similar characteristics, being more specific regarding our study aims and the analyses performed. For instance, in the first paragraph of the discussion section, we have added the following sentence:

“The major contributors of inequality were the wealth index, higher education and living in an urban setting.”

20. Line 260. Please elaborate what kind of ‘unhealthy routine behaviours’ does the author meant here.

Answer. We have added the following information: “...unhealthy routine behaviors, such as sedentarism and consumption of high caloric food..”. The explanation of these behaviors in the aforementioned paragraph has not been expanded, because they are explained in the following paragraphs of the discussion section. 

21. Line 261. Please also elaborate on what is the ‘ultra-processed food’.

Answer. Thank you for the observation. Ultra-processed food is a category of high caloric food, which passes through multiple processes usually including the addition of many ingredients such as salt, sugar and oil as defined by the NOVA classification proposed by the Food and Agriculture Organization of the United Nations [1]. Examples of this food are soft drinks, chocolate, ice-cream, chips, hotdogs, fries, etc. In order to avoid confusion, we have modified the term in the sentence cited.

1. Monteiro C.A, Cannon G, Lawrence M, Costa M.L, Pereira P. Ultra-processed foods,

diet quality, and health using the NOVA classification system. Food and Agriculture Organization of the United Nations. Rome; 2019. 

Now it reads: “...in addition, these people are exposed to a great amount of advertising of caloric dense food” 

22. Line 268, 270, 273, 281. Please mention in which country is the cited studies were conducted? This will help the reader understand the context.

Answer We agree with the reviewer on this point. We have added the countries according to the bibliography to support the manuscript and make it easier for readers to understand. 

23. Line 280. “The population in urban areas is 1.9 times more likely to have lower physical activity when compared to that living in rural areas [42].” If this is not the findings of this study, please add detailed information on where is the reference study was conducted etc.

Answer. Thank you for the observation, as mentioned above we have specified the countries according to the bibliography. 

24. Line 281. “It has been reported that occupational physical activity in metropolitan areas has been reduced by about 120 kcal/day [45].” Where it is reported? Who reported it?

Answer. We thank the reviewer for bringing this to our attention. We found this information in the journal “Nutrición Hospitalaria”; however, this study did not have these data as their findings. Therefore, we have modified this reference to one reporting that in the last 50 years Americans have decreased physical activity and that the public health measures implemented are still not insufficient to stop this trend. 

Reference 45 is now number 48: 

● Church TS, Thomas DM, Tudor-Locke C, Katzmarzyk PT, Earnest CP, Rodarte RQ, et al. Trends over 5 decades in U.S. occupation-related physical activity and their associations with obesity. PLoS One. 2011;6(5):e19657.

25. Line 287-289. “Roads in rural areas are usually very rough with insufficient access ways, thus residents of this area are conditioned to do more physical activity as part of their daily commute [47]”

This sentence cites WHO report on ageing and health, is that correct?

Answer. We appreciate the reviewer's comment. This was a mistake and the reference mentioned has been replaced by the correct one. 

Reference 47 is now number 50: 

● Day K. Physical environment correlates of physical activity in developing countries: A review. J Phys Act Health. 2018;15(4):303–14.

26. Line 296-298. “This was concentrated in the 30-59 age group, a fact that could be explained both by the hormonal component typical of menopause and postmenopause [50],”

What does ‘This’ refer to? I think it is better to rephrase to “Among women, abdominal obesity was….”.

I also do not agree to connect this age group to menopause or post menopause. How does the author explain those women in the age group 60 or more having lower prevalence of AO? On the opposites, in the next sentences, the author mentioned about the fertility rate in the same age group.

Answer. We thank the reviewer for bringing this to our attention. By “this” we refer to the higher prevalence of AO in women. We have changed the first paragraph to make this clearer and reference 50 about menopause and postmenopause has been deleted. Regarding the fertility rate, we mean that young women (18-29 years old), who have more AO, compared to their male counterparts, could be explained by the higher fertility rate in the group of 20-34 years old in Perú. 

It now reads: “The prevalence of AO was greatest among women belonging to the 30-59 age group, which might be explained by the decline in regular exercise and the use of diets over time [18].”

27. Line 200-300. “We must acknowledge the large proportion of young women with AO compared to their male counterparts”.

Do ‘young women’ here refer to women in the age group 18-29 years old?

Answer. Thank you for bringing this to our attention. Yes, by “Young women“ we mean the age group from 18 to 29 years old, in whom the prevalence of AO was 70.8% compared to males presenting a prevalence of 36.1%, with a 34.7 percentage point difference. We have modified the cited text to clarify.

It now reads: “Of note was the large proportion of women aged 18-29 years of age with AO compared to their male counterparts. ”

28. Line 301. “In Peru, women between 20-34 years of age reach the highest levels of fertility, exceeding 60% in both rural and urban areas [51],”

What is exceeding 60%? What is this number refer to?

Answer. Thank you for bringing this to our attention, we meant to stress the fact that women in this age group have the highest prevalence of fertility both in rural and urban areas. We have modified the sentence.

It now reads: “In Peru, the prevalence of fertility in women between 20-34 years of age exceeds 60% in both rural and urban areas [53]”

29. Line 302-308. Table 1 showed that the majority of women in this study were having secondary (37%) and higher (33.5%) education level. Which means the participation of Peruvian women in the labour market may be higher. It somehow contradicts the two sentences here. How do you explain this?

Answer. Despite a large proportion of residents of both sexes in Peru being able to obtain secondary and higher education, the rate of participation of women in the workforce was approximately 16 percentage points less than that of men in 2018 [1]. Even among those with higher education it is estimated that 86.2% of men are employed vs. 74.5% of women. Likewise, the average monthly salary of men (1846 PEN (487.69 USD)) is higher than that obtained by women (1322 PEN (349.26 USD)) [2]. These differences might induce women to leave behind their professional aspirations in favor of raising their children and tending to other household chores. 

1. INEI. [Peru: Gender Gaps 2016. Progress towards equality between women and men]. Inst Nac Estadística e Informática. 2016; 

2.Instituto Nacional de Estadistica e Informatica. [Technical Report: Statistics with a gender perspective - Trimester: October-November-December 2019] [Internet]. 2020. Available from: https://www.inei.gob.pe/media/MenuRecursivo/boletines/01-informe-tecnico-n01_estadisticas-genero_oct-nov-dic2019.PDF

To further illustrate this point we have added a few sentences in the referenced paragraph: “Even among individuals with higher a education it is estimated that 86.2% of men in Perú are employed vs. 74.5% of women. This percentage drops to 67.8% among women with a primary education but remains around 82.2% for their male counterparts. Likewise, the average monthly salary of men of 1846 PEN (487.69 USD) is higher than the 1322 PEN (349.26 USD) earned by women with [55]. Therefore, Peruvian women may be less motivated to pursue their professional career path due to inequalities in the workforce and a lack of monetary incentive;... ”

We have added one reference:

55. Instituto Nacional de Estadistica e Informatica. [Technical Report: Statistics with a gender perspective - Trimester: October-November-December 2019] [Internet]. 2020. Available from: https://www.inei.gob.pe/media/MenuRecursivo/boletines/01-informe-tecnico-n01_estadisticas-genero_oct-nov-dic2019.PDF

30. Line 308-309. “Finally, due to social beliefs in some contexts, obesity could be seen as a symbol of good economic status”, Which contexts? Does this social belief also exist in Peru?

Answer. In the social context of Indonesia cultural factors and social beliefs might influence the burden of obesity. Indeed, like this country, the same social trend may occur in Peru. An example of this is a study conducted in an urban Peruvian population in Lima - Peru which found that obesity was a source of pride because it is perceived to be associated with a better economic position in society.

One reference has been added as number 59 in the text:

59. Sifuentes- León E, Rivas-Díaz L. [ Obesity and Overweight in Beliefs and Attitudes of Residents an Urban Community From the Sociology of Health]. Investigaciones sociales. 2019; 22(41):261-277.

31. Line 308-311. “Finally, due to social beliefs in some contexts, obesity could be seen as a symbol of good economic status [19], becoming widely accepted among women because of its association with pregnancy and child-bearing [55].”

This sentence should be divided into two sentences, as it talked about obesity from a different point i.e., obesity as a symbol of good economic status and obesity association with pregnancy and childbearing.

Answer. We agree with your observation. The two sentences were separated according to the suggestion and the order of the sentences has also been changed for better linkage. 

It now reads: “... being widely accepted among women because of its association with pregnancy and child-bearing [58]. Due to social beliefs in countries such as Indonesia and Perú [59], obesity might be seen as a symbol of good economic status [18]. ...”

32. Line 325. “We found that the socioeconomic characteristics of the population impact the distribution of AO in Peru”.

The author can delete ‘We found that’ in this conclusion section.

The socioeconomic inequality in AO exists, but it is less prominent among women.

Answer. We agree with your observation and have changed the sentence to make it clearer. 

It now reads: “The socioeconomic characteristics of the population impact the distribution of AO in Peru. This inequality is more prominent among men although AO is more prevalent among women. …”

33. Reference.

Perhaps the author should update some outdated references to more recent publications.

Several references were not in English, and no translation made available in the Reference section. Some references were not complete and not cited according to Vancouver style. 

Answer. We have reviewed our references and updated those related to associations made between abdominal obesity and various risk and sociodemographic factors where pertinent. References in Spanish have been translated to English by the authors. We must emphasize that some of the older references pertain to various economic and statistical measures and analyses that were published over a decade ago and updated data is limited. 

The authors.

---

## [Decision Letter · Decision Letter 1]

25 Jun 2021

Socioeconomic inequalities in abdominal obesity among Peruvian adults

PONE-D-20-35836R1

Dear Dr. Barrenechea-Pulache,

We’re pleased to inform you that your manuscript has been judged scientifically suitable for publication and will be formally accepted for publication once it meets all outstanding technical requirements.

Kind regards,

Isil Ergin, Assoc. Prof.

Academic Editor

PLOS ONE

Additional Editor Comments (optional):

Reviewers' comments:

Reviewer's Responses to Questions

**Comments to the Author**

1. If the authors have adequately addressed your comments raised in a previous round of review and you feel that this manuscript is now acceptable for publication, you may indicate that here to bypass the “Comments to the Author” section, enter your conflict of interest statement in the “Confidential to Editor” section, and submit your "Accept" recommendation.

Reviewer #1: All comments have been addressed

2. Is the manuscript technically sound, and do the data support the conclusions?

Reviewer #1: (No Response)

3. Has the statistical analysis been performed appropriately and rigorously? 

Reviewer #1: (No Response)

4. Have the authors made all data underlying the findings in their manuscript fully available?

Reviewer #1: (No Response)

5. Is the manuscript presented in an intelligible fashion and written in standard English?

Reviewer #1: (No Response)

6. Review Comments to the Author

Reviewer #1: (No Response)

7. PLOS authors have the option to publish the peer review history of their article (what does this mean?). If published, this will include your full peer review and any attached files.

Reviewer #1: No

---

## [Editor Report · Acceptance letter]

30 Jun 2021

PONE-D-20-35836R1 

Socioeconomic inequalities in abdominal obesity among Peruvian adults 

Dear Dr. Barrenechea-Pulache:

I'm pleased to inform you that your manuscript has been deemed suitable for publication in PLOS ONE. Congratulations! Your manuscript is now with our production department. 

Kind regards, 

on behalf of

Dr. Isil Ergin 

Academic Editor

PLOS ONE